# Bearing Fault Diagnosis Based on Randomized Fisher Discriminant Analysis

**DOI:** 10.3390/s22218093

**Published:** 2022-10-22

**Authors:** Hejun Ye, Ping Wu, Yifei Huo, Xuemei Wang, Yuchen He, Xujie Zhang, Jinfeng Gao

**Affiliations:** 1School of Information Science and Engineering, Zhejiang Sci-Tech University, Hangzhou 310018, China; 2Key Laboratory of Intelligent Manufacturing Quality Big Data Tracing and Analysis of Zhejiang Province, China Jiliang University, Hangzhou 310018, China; 3College of Control Science and Engineering, Zhejiang University, Hangzhou 310058, China

**Keywords:** bearing, fault diagnosis, random Fourier feature, Fisher discriminant analysis

## Abstract

In this paper, a novel randomized Fisher discriminant analysis (RFDA) based bearing fault diagnosis method is proposed. First, several representative time-domain features are extracted from the raw vibration signals. Second, linear Fisher discriminant analysis (FDA) is extended to nonlinear FDA named RFDA by introducing the random feature map to deal with the non-linearity issue. Specifically, the extracted time-domain features data are mapped onto a high-dimensional space using the random feature map function rather than kernel functions. Third, the time-domain features are fed into the built RFDA model to extract the discriminant features for diagnosis. Moreover, a Bayesian inference is employed to identify the class of the collected vibration signals to diagnose the bearing status. The proposed method uses random Fourier features to approximate the kernel matrix in the kernel Fisher discriminant analysis. Through employing randomized Fisher discriminant analysis, the nonlinearity issue is dealt with, and the computational burden is remarkably reduced compared to the kernel Fisher discriminant analysis (KFDA). To illustrate the superior performance of the proposed RFDA-based bearing fault diagnosis method, comparative experiments are conducted on two widely used datasets, the Case Western Reserve University (CWRU) bearing dataset and the Paderborn University (PU) bearing dataset. For the CWRU dataset, the computation time of RFDA is much shorter than KFDA, while the accuracy rate reaches the same level of KFDA. For the PU dataset, the accuracy rate of RFDA is slightly higher than KFDA, and the computation time is only 44.14% of KFDA.

## 1. Introduction

Bearings are an essential and key part that is widely used in modern rotating machinery. As a vital component, the occurrence of bearing faults will result in significant breakdown time, increasing maintenance costs, and even jeopardizing casualties. Therefore, it is critical to precisely and quickly diagnose the bearing status [1,2,3]. To explore the bearing status, a variety of signals are collected and used, such as acoustic signals [4], vibration signals [5], and current signals [6]. Among them, the vibration signals contain abundant fault energy information, and the data acquisition of bearing vibration signals does not require complex equipment and professionals.

Therefore, the vibration signal is popularly used to monitor the bearing status. Bearing fault diagnosis techniques via vibration signals can be generally categorized into two classes, including signal-analysis-based and data-driven methods. Regarding the signal-analysis-based method, the raw vibration signals are firstly analyzed using signal processing methods such as time-domain analysis [7,8], frequency-domain analysis [9] and time–frequency-domain analysis [10]. Afterward, the bearing status is determined by features extracted from different domains using expert knowledge.

Data-driven methods depend only on the vibration signals, as opposed to signal analysis-based methods. In data-driven methods, the bearing fault diagnosis becomes a pattern recognition problem. To deal with the pattern recognition of high-dimensional data in bearing fault diagnosis, dimensionality reduction techniques have been widely employed, such as principal component analysis (PCA) [11], locality preserving projection (LPP) [12], and recurrence analysis (RA) [13,14]. Moreover, t-distributed stochastic neighbor embedding (t-SNE) is an efficient dimensionality reduction tool. t-SNE identifies close similarities between samples through the relative location of points in the mapped feature space. For t-SNE, the number of features in the reduced space is not restricted by the number of output dimensions [15]. In [15], t-SNE is combined with the multiscale distribution entropy method to extract the low-dimensional nonlinear complexity features from the vibration signals of rolling bearing. Similarly, uniform manifold approximation and projection (UMAP) uses graph layout algorithms to arrange data in a low-dimensional space [16]. In [16], UMAP is combined with feature selection techniques to improve the performance of latent space visualization for chemical process data. As a popular unsupervised learning method, autoencoder (AE) can learn effective features with unlabeled data by minimizing the error between original input and reconstructed input. Deep autoencoder (DAE) has been widely used to extract hierarchical features for bearing fault diagnosis from vibration signal [17]. Variational autoencoder (VAE) generates a latent representation of the data through imposing a distribution over the latent variables [18]. In [18], a semi-supervised learning scheme using variational autoencoder (VAE)-based deep generative models was proposed to deal with the small labeled data problem in the bearing fault diagnosis.

Compared to unsupervised dimensionality reduction techniques, the supervised method, Fisher discriminant analysis (FDA), aims to find the low-dimensional representation from the high-dimensional data to simultaneously maximize the distance between different classes and minimize the distance within the same class. FDA was also named linear discriminant analysis (LDA). More importantly, FDA can fully utilize the labeled information to directly offer the classification results. Therefore, FDA has gained considerable attention to achieving the task of bearing fault diagnosis in recent years. Due to its simplicity and efficiency, FDA has proved its superiority in fault diagnosis.

Jin et al. [19] developed trace ratio linear discriminant analysis (TR-LDA) to address the non-Gaussian data by solving the trace ratio problem. Zhou et al. [20] employed LDA to reduce the dimensionality of 10 statistical features of the raw vibration signals and its transient component using transient-extracting transform (TET) for improving the fault diagnosis performance. To address the nonlinearity issue contained in vibration signals, variants of nonlinear fisher discriminant analysis have been proposed. One of the most frequent extensions is kernel Fisher discriminant analysis (KFDA) [21]. The idea behind KFDA is to map the original data onto a high-dimensional feature space in which linearly separable feature space is expected for further classification. Van et al. [22] presented a wavelet kernel-based local Fisher discriminant analysis (WKLFDA) to extract the nonlinear features from original vibration signals for bearing fault diagnosis. Jiang et al. [23] proposed a semi-supervised kernel marginal Fisher analysis (SSKMFA) method to investigate the inherent manifold structure embedded in data, and simultaneously took the intra-class compactness and the inter-class separability into account. Tao et al. [24] developed a semi-supervised kernel local Fisher discriminant analysis (SSKLFDA) through introducing the regularization term with pseudo labels by utilizing unlabeled data for the supervised dimensionality reduction.

However, a large number of features extracted from kernel methods will increase the computation burden and may lead to poor fault diagnosis performance, due to all training data being involved in the calculation of the kernel matrix. To lessen the computing complexity of KFDA, Li et al. [25] presented a feature vector selection (FVS) approach by using a subset of the samples to present all of the data under the geometrical consideration. Liu et al. [26] combined the kernel feature selection method and KFDA technique to reduce the computation burden and alleviate the impact of irrelevant features in fault diagnosis. Recently, random feature map was widely studied in large-scale kernel machines [27,28]. Rahimi and Recht suggested a random Fourier feature mapping approach for approximating non-linear kernels by mapping the input data to randomized low-dimensional feature space [27]. Fisher discriminant analysis using random Fourier feature mapping was developed to accelerate kernel Fisher discriminant analysis in [29,30]. In [30], a random feature map was introduced to map the input data to a finite dimension to accelerate FDA and kernel FDA. Moreover, a theoretical guarantee was offered to prove that the FDA algorithms using random projection can derive good generalization ability. In [29], the randomized solution to linear discriminant analysis was developed for processing hyperspectral images to overcome the dimensionality problem. To the best of our knowledge, there is little investigation of Fisher discriminant analysis using random feature map in bearing fault diagnostics.

Motivated by the above discussions, a new bearing fault diagnosis method is proposed by using randomized Fisher discriminant analysis (RFDA). Specifically, 12 time-domain features are first extracted from original vibration signals. Then, an RFDA model is built by using the extracted 12 time-domain features for fault diagnosis. In RFDA, the high-dimensional data are mapped to a low-dimensional features space using random feature mapping, then the projection matrix with Fisher discriminant analysis is calculated. To identify the state of bearings, a Bayesian inference is employed. The main contributions of this paper lie on the following aspects:RFDA, a nonlinear variant of FDA, is utilized for bearing fault diagnosis. The RFDA-based method can achieve similar performance to the KFDA-based method, while the computational burden is remarkably reduced.Two widely used bearing datasets are employed to validate the effectiveness of the proposed RFDA-based bearing fault diagnosis method. Results show the superior performance of the proposed method over other related methods.

The remainder of this paper is structured as follows. In Section 2, a brief review of Fisher discriminant analysis and random Fourier feature map is given. In Section 3, details of the proposed RFDA based fault diagnosis method are described. In Section 4, two experimental datasets of bearings are used to assess the performance of the proposed method comparing with other related methods. Conclusions are drawn in the final Section 5.

## 2. Related Works

### 2.1. Fisher Discriminant Analysis

The aim of the FDA is to find a linear transformation matrix to separate the projections in the low-dimensional space as much as possible. For this purpose, FDA measures the compactness of each class with the within-class scatter matrix and the distance between classes with the between-class scatter matrix. Through maximizing the ratio of between-class to within-class scatter matrices, the optimal transformation matrix is calculated.

Denote the training dataset as *D*,
$$D=\left\{\left(x_{1},y_{1}\right),\left(x_{2},y_{2}\right),\ldots ,\left(x_{N},y_{N}\right)\right\}$$
where the sample $x_{i}\in R^{m}$ is the *m*-dimensional vector. $y_{i}$ represents the category of $x_{i}$. *N* samples are contained in the training dataset *D*.

Assumed that there are *k* types in *D*. Define $n_{j}\left(j=1,2,\cdots ,c\right)$ as the number of samples of type $y_{j}$, and $X_{j}\left(j=1,2,\cdots ,c\right)$ as the set of all samples which belong to type $y_{j}$. Then, the between-class scatter matrix $S_{b}$ is defined as
(1)$$S_{b}=\sum_{j=1}^{c}n_{j}\left(\mu_{j}-\mu \right){\left(\mu_{j}-\mu \right)}^{T}$$
where $\mu_{j}\left(j=1,2,\cdots ,c\right)$ is the mean vector of the type $y_{j}$, and μ is the mean vector of all samples.

The within-class scatter matrix $S_{w}$ is defined as,
(2)$$S_{w}=\sum_{j=1}^{c}\sum_{x\in X_{j}}\left(x-\mu_{j}\right){\left(x-\mu_{j}\right)}^{T}$$

To find the optimal projection matrix W, the optimization problem of FDA is defined as,
(3)$$\arg\max_{W}J_{FDA}=\frac{\mathrm{Tr}\left(W^{T}S_{b}W\right)}{\mathrm{Tr}\left(W^{T}S_{w}W\right)}$$
where Tr(·) is the trace operator.

Assumed that $S_{w}$ is non-singular, the solution of the optimization problem (3) can be feasibly obtained by using generalized eigenvalue decomposition [31],
(4)$$S_{b}w_{j}=\rho_{j}S_{w}w_{j}$$
where $w_{j}\in R^{m}$ is the eigenvector and the corresponding eigenvalue is $\rho_{j}$. Generally, the eigenvectors which are corresponding to the largest *d* eigenvalues are retained for the purpose of dimensionality reduction. Thus, the projection matrix $\bar{W}$ is constructed as $\bar{W}=\left[\begin{array}{cccc} w_{1} & w_{2} & \cdots & w_{d} \end{array}\right]\in R^{m\times d}$.

Using the derived projection matrix $\bar{W}$, the low-dimensional projection vector $z_{i}\in R^{d}$ of the original data $x_{i}$ is,
(5)$$z_{i}=x_{i}^{T}\bar{W}$$

### 2.2. Random Fourier Feature Map

To address the issue of nonlinearity, kernel methods are usually employed to extend the linear dimensionality reduction methods to their nonlinear variants. According to Cover’s theorem, the original input data $x\in R^{m}$ is mapped to a high-dimensional, even an infinite, reproducing kernel Hilbert space (RKHS) $\in R^{F}$ by a given nonlinear mapping function ϕ.

Define the mapping function as,
(6)$$\phi \left(x\right):R^{m}\to R^{F}$$

Since the nonlinear features ϕ(x) are created from the implicit mapping function, the core of the kernel methods relies on the implicit lifting through the kernel trick. Using the kernel trick, the nonlinear features are generated by calculating the inner product between pairs of input points.

The inner product between lifted data points $\phi (x_{i})$ and $\phi (x_{j})$ is defined as,
(7)$$k\left(x_{i},x_{j}\right)=\langle \phi \left(x_{i}\right),\phi \left(x_{j}\right)\rangle$$
where 〈·〉 represents the inner product operator.

In the kernel method, the kernel matrix can be constructed for *N* training samples $\left\{\left.x_{1},x_{2},\ldots ,x_{N}\right.\right\}$ as,
$$K={\left[k\left(x_{i},x_{j}\right)\right]}_{i,j=1,2,\cdots ,N}$$

To generate the features for a testing data point, the kernel matrix K should be evaluated. As displayed in the kernel matrix K, all training data are involved. As a result, the disadvantage of the kernel method is that there are large computational and storage costs while facing large training sets.

For addressing this issue, the shift-invariant kernels $k(x_{i},x_{j})$ are related to random nonlinear features with the help of Bochner’s theorem [32]. Defining $\beta_{\omega }\left(x\right)=exp\left(-j\omega^{T}\left(x\right)\right)$, the inner product $k\left(x_{i},x_{k}\right)$ can be expressed as,
(8)$$\begin{array}{cc} k\left(x_{i},x_{k}\right) & =\int_{R^{d}}p\left(\omega \right)exp\left(-j\omega^{T}\left(x_{i}-x_{k}\right)\right)d\omega \\ \; & =E_{\omega }\left[\beta_{\omega }\left(x_{i}\right)\beta_{\omega }{\left(x_{k}\right)}^{*}\right] \end{array}$$
where p(ω) is the inverse Fourier transform of *k*. ω is sampled from the distribution p(ω). $E_{\omega }\left[\cdot \right]$ is the expectation operator. $\beta_{\omega }{\left(x_{k}\right)}^{*}$ is the complex conjugate of the inverse Fourier transform of $\beta_{\omega }\left(x_{k}\right)$. $\beta_{\omega }\left(x_{i}\right)\beta_{\omega }{\left(x_{k}\right)}^{*}$ can be considered an unbiased estimate of $k(x_{i},x_{k})$.

Furthermore, the kernel $k(x_{i},x_{k})$ is approximated as below,
(9)$$\begin{array}{cc} k(x_{i},x_{k}) & =\int_{R^{d}}p\left(\omega \right)exp\left(-j\omega^{T}\left(x_{i}-x_{k}\right)\right)d\omega \\ \; & =E_{\omega }\left[\beta_{\omega }\left(x_{i}\right)\beta_{\omega }{\left(x_{k}\right)}^{*}\right]\\ \; & \approx \sum_{j=1}^{D}\frac{1}{D}exp\left(-j\omega_{j}^{T}x_{i}\right)exp\left(-j\omega_{j}^{T}x_{k}\right) \end{array}$$
where D≪N.

According to Euler’s formula, it obtains
(10)$$exp\left(-j\omega^{T}x\right)=cos\left(\omega x\right)+jsin\left(\omega x\right)$$

To obtain a real-value random feature for *k*, the distribution p(ω) and kernel *k* should be real. Thus, only the real part of the exponential would remain in Equation (Equation 11). By replacing $exp(-j\omega^{T}\left(x_{i}-x_{k}\right))$ with $cos(\omega^{T}\left(x_{i}-x_{k}\right))$ [27], then
(11)$$\begin{array}{cc} k(x_{i},x_{k}) & \approx \sum_{j=1}^{D}\frac{1}{D}exp\left(-j\omega_{j}^{T}x_{i}\right)exp\left(-j\omega_{j}^{T}x_{k}\right)\\ \; & =\langle \frac{1}{\sqrt{D}}z\left(x_{i}\right),\frac{1}{\sqrt{D}}z\left(x_{k}\right)\rangle \end{array}$$

Based on Equation (Equation 11), the original data are explicitly projected onto a low-dimensional Euclidean inner product space through randomized feature map in [27], instead of implicitly mapping function ϕ. Following this idea, the inner product $k(x_{i},x_{j})$ is approximated by defining an explicit random feature map function $z\left(x\right):R^{m}\to R^{D}$, where
(12)$$k\left(x_{i},x_{j}\right)=\langle \phi \left(x_{i}\right),\phi \left(x_{j}\right)\rangle \approx z{\left(x_{i}\right)}^{T}z\left(x_{j}\right)$$
where $z_{\omega }\left(x\right)=\sqrt{2}cos\left(\omega^{T}x+b\right)$ with *b* is drawn from uniform distribution U[0,2π]. It is noted that the parameters *b* are used to make the expectation of the inner product of $z_{\omega }\left(x\right)$ close to the shift-invariant kernel. For further explanation of the parameters *b*, refer to [27,33].

Thus, the inner product $z_{\omega }{\left(x_{i}\right)}^{T}z_{\omega }\left(x_{k}\right)$ is expressed as $z_{\omega }{\left(x_{i}\right)}^{T}z_{\omega }\left(x_{k}\right)=\frac{1}{D}\sum_{j=1}^{D}z_{\omega_{j}}\left(x_{i}\right)$

$z_{\omega_{j}}\left(x_{k}\right)$ where D samples of $z_{\omega_{j}}$ are randomly chosen. Define D-dimensional vector $z_{\omega }\left(x\right)$
(13)$$z_{\omega }\left(x\right)=\sqrt{\frac{2}{D}}\left[\begin{array}{c} cos(\omega_{1}^{T}x+b_{1})\\ cos(\omega_{2}^{T}x+b_{2})\\ \vdots \\ cos(\omega_{D}^{T}x+b_{D}) \end{array}\right]$$

Based on the random feature map z, a D-dimensional feature is obtained from the original data x. Then, linear FDA is conducted in the D-dimensional feature space. The details of Fisher discriminant analysis with random feature map are elaborated in the next section.

**Remark** **1.**
*The bound error between the kernel matrix $K_{i,k}=k\left(x_{i},x_{k}\right)$ and ${\bar{K}}_{i,k}=z{\left(x_{i}\right)}^{T}z\left(x_{k}\right)$ was discovered in spectral norm as in [28],*

(14)
$$E∥K-\bar{K}∥\leq \sqrt{\frac{3N^{2}\log N}{D}}+\frac{2N\log N}{D}$$


*Obviously, the error becomes small as the dimensionality D is selected as a large number. Yet, the computational cost will also increase. To balance the computational burden and the error, the dimensionality D of random features can be determined by cross-validation in the offline training phase.*


## 3. RFDA-Based Fault Diagnosis

The RFDA-based fault diagnosis is developed in this section. First, the time-domain features are extracted from the collected raw vibration signals. Then, an RFDA model is trained using these time-domain features. To identify the class of vibration signals, a Bayesian inference is employed.

### 3.1. Time-Domain Feature Extraction

Time-domain statistical features, such as the impulsive factor, kurtosis, skewness, peak factor, and root mean square, are usually employed to detect and identify the bearing damage. In this study, 12 time-domain features are computed from each signal to exhibit different distribution characteristics of the raw vibration signals. Table 1 lists the detailed descriptions of the adopted time-domain features.

From Table 1, the time-domain features of the raw vibration signals $x=\{x_{1},x_{2},\cdots ,x_{n}\}$ are extracted and denoted as $h=f\left(x\right)={\left\{x_{p},x_{pp},\bar{x},\left|\bar{x}\right|,x_{sra},x_{v},x_{std},x_{rms},x_{k},x_{ske},x_{pf},x_{if}\right\}}^{T}\in R^{12}$ for the RFDA modeling, where f(·) is the function of the time-domain feature extraction.

### 3.2. RFDA Model Training

Assumed that $N_{all}$ samples are collected from fault-free and c−1 faulty classes, where each sample has *n* data points, then $N_{all}$h are computed according to Table 1.

For the purpose of training the RFDA model, the extracted time-domain features h are normalized,
(15)$$\hat{h}=\frac{h-h_{mean}}{h_{std}}$$
where $h_{mean}$ and $h_{std}$ are the mean and standard variance of the time-domain features h extracted from normal data, respectively.

To address the issue of nonlinearity in data, the random Fourier feature of h is extracted using random feature map,
(16)$$z\left(\hat{h}\right)=\sqrt{\frac{2}{m}}\left[\begin{array}{c} \cos\left(\omega_{1}^{T}\hat{h}+b_{1}\right)\\ \vdots \\ \cos\left(\omega_{m}^{T}\hat{h}+b_{m}\right) \end{array}\right]\in R^{m}$$
where *m* is the dimension of random features.

In kernel-based fault diagnosis, the Gaussian radial basis function (RBF) kernel is usually employed due to its generalization ability. The RBF kernel is defined as,
(17)$$K\left(\delta \right)=exp\left(-\frac{{∥\delta ∥}_{2}^{2}}{s}\right)$$
where *s* is the kernel width of RBF kernel. Corresponding to the selected RBF kernel, the parameter ω of random Fourier features $z(\hat{h})$ is sampled from the following Gaussian distribution,
(18)ω∼N(0,2I/s)
where I is the identity matrix with appropriate dimension.

Then, using Equations (15)–(17) random Fourier features, the between-class scatter matrix ($S_{hb}$) and within-class scatter matrix ($S_{hw}$) are formed,
(19)$$S_{hb}=\sum_{j=1}^{c}N_{j}\left(\hat{z_{h_{m,j}}}-\hat{z_{h_{m}}}\right){\left(\hat{z_{h_{m,j}}}-\hat{z_{h_{m}}}\right)}^{T}$$
where $\hat{z_{h_{m,j}}}$ is the mean of random Fourier features, which belong to the j=1,⋯,cth class. $\hat{z_{h_{m}}}$ is the mean of all extracted features.
(20)$$S_{hw}=\sum_{j=1}^{c}S_{hwj}=\sum_{j=1}^{c}\sum_{\hat{z_{h}}\in {\hat{H}}_{j}}\left(\hat{z_{h}}-\hat{z_{h_{m,j}}}\right){\left(\hat{z_{h}}-\hat{z_{h_{m,j}}}\right)}^{T}$$
where ${\hat{H}}_{j}$ represents the domain of the *j*th class.

Similar to Equation (Equation 4), to seek the projection matrix $W_{h}$ that maximizes the distance between samples that belong to different classes and minimize the distance between samples that belong to the same classes, the following generalized eigenvalue problem is to be solved:(21)$$S_{hb}w_{hi}=\lambda_{i}S_{hw}w_{hi}$$
where $\lambda_{i}$ is the eigenvalue corresponding to the eigenvector $w_{hi}$.

Using eigenvalue decomposition, the solution of Equation (Equation 20) can be derived. To reduce the dimensionality, only eigenvectors corresponding to large eigenvalues are retained. Thus, the projection matrix ${\bar{W}}_{h}$ is formed as ${\bar{W}}_{h}=\left[\begin{array}{cccc} w_{h1} & w_{h2} & \cdots & w_{hd} \end{array}\right]\in R^{m\times d}$, where *d* is the number of retained latent variables.

**Remark** **2.**
*The selection of the number of retained latent variables d can be determined by minimizing the information criterion similar to Akaike’s information criterion (AIC) [34]. In discriminant analysis, the number of retained latent variables d is usually set as c−1, where c is the number of sample classes.*


### 3.3. RFDA-Based Bearing Fault Diagnosis Scheme

Through the projection matrix ${\bar{W}}_{h}$ derived from the RFDA model, the projection $d_{ht}$ of a new vibration sample $x_{t}$ can be calculated as,
(22)$$d_{ht}={\bar{W}}_{h}^{T}z\left(f\left(x_{t}\right)\right)$$

Based on the established RFDA model, k-nearest neighbors (kNN) and Bayesian inference are often adopted to classify the class of the new samples. Due to its simplicity and low computational burden, Bayesian inference is used in this study. Bayesian inference is based on the posterior probability to determine the class of new data samples. Assuming that the projections $d_{ht}$ are Gaussian distributed, then the class membership can be determined by the Fisher discriminant function, which is defined as [35],
(23)$$\begin{array}{cc} g_{j}\left(d_{ht}\right)= & -\frac{1}{2}[{\left(d_{ht}-d_{m,j}\right)}^{T}{\left(\frac{1}{n_{j}-1}W_{n}^{T}S_{hw,j}W_{h}\right)}^{-1}\left(d_{ht}-d_{m,j}\right)\\ & -\ln[\det\left(\frac{1}{n_{j}-1}W_{h}^{T}S_{hw,j}W_{b}\right)]] \end{array}$$
where $d_{m,j}=W_{b}^{T}{\hat{h}}_{m,j}$.

In accordance with the Fisher discriminant function, the class of the test data sample $x_{t}$ is classified to the class, where the value of the Fisher discriminant function is the largest:(24)$$C\left(x_{t}\right)=arg\max_{1\leq j\leq c}g_{j}\left({\bar{W}}_{h}^{T}z\left(f\left(x_{t}\right)\right)\right)$$

The flowchart of the proposed RFDA-based bearing diagnosis is depicted in Figure 1.

## 4. Experiments and Results

In this section, to demonstrate the applicability and effectiveness of the proposed RFDA-based fault diagnosis method, experiments on two popular bearing benchmark datasets are conducted. For comparison study, FDA and KFDA [36] are used. The simulation environment is Intel Core i7-8750h CPU@2.20 GHz CPU and 24 GB RAM running under Windows 10 with MATLAB R2020b.

### 4.1. Case 1: CWRU Dataset

As a commonly and widely used bearing dataset for the evaluation of fault diagnosis performance, the CWRU datasets were collected and provided by the Case Western Reserve University bearing data center on the experimental facility shown in Figure 2.

In the CWRU dataset, the data with the acquisition frequency of 12 K are selected, and several single-point faults are introduced on the outer ring, the inner ring, and rolling element, with fault diameters of 0.007 for each position. For each scenario, 50 samples are collected. A sample is composed of 1024-point vibration sequences without overlap. The number of training and testing datasets and the descriptions of fault scenarios are shown in Table 2.

To assess the fault diagnosis performance quantitatively, the widely used index (i.e., accuracy rate) is adopted in this work. The definition of accuracy rate is
(25)$$Acc=\frac{TN+TP}{TN+FN+FP+TP}$$
where TN, FN, FP and TP denote the number of true negative, false negative, false positive, and true positive outcomes, respectively.

To determine the kernel width of KFDA, the cross-validation method is used. Figure 3 shows the fault classification results of the KFDA model with different kernel widths. From Figure 3, the optimal kernel width is selected as 700. The computation time is kept stable (i.e., 0.24 s). To make the comparison fair, the same kernel width *s* is used for RFDA. Since the selection of dimension *m* not only affects the performance of approximation of kernel matrix, but also requires different computational burdens in RFDA-based fault diagnosis method. To obtain optimal parameters of RFDA, the accuracy rates with different parameters are plotted in Figure 4. We can set s=700,m=50 from Figure 4. The number of maintained latent variables is selected as 3 for all methods.

To visualize the features vividly, the extracted features are displayed in Figure 5. From Figure 5, it can be observed that the distances between different clusters corresponding to specific scenarios are large. Compared to FDA, RFDA and KFDA can provide more efficient discriminant performance, since the distance between the data of the same category is smaller.

The accuracy rates and computation times of FDA, KFDA, and RFDA using the testing dataset are given in Table 3. In Table 3, the accuracy rate and computation time are the average results of 100 experiments. From the data in Table 3, the accuracy rates of FDA, RFDA, and KFDA can achieve 100%. The reason is that the magnitudes of the faults are large in the CWRU data, making them easily classified. On the other hand, the average computation time of RFDA is 0.1232 s, which is at the same level as FDA. However, the computation time of KFDA is almost two times that of RFDA. Thus, the computation burden of the calculation of kernel matrix is hugely reduced in RFDA through the adoption of the random Fourier feature.

### 4.2. Case 2: PU Dataset

PU dataset is another popular bearing dataset provided by the Paderborn University Bearing Data Center for comparing the fault diagnosis performance [37]. The test rig of PU datasets is displayed in Figure 6.

There are 14 types of real failure data in PU dataset. For details of the working conditions, refer to [37]. In this study, the vibration data were collected under the working conditions of rotating speed 1500 rpm, load torque 0.7 nm, and radial force 1000 N. The sampling frequency of the vibration data is 64 KHz, and the sampling time is 4 s. In the training dataset, 100 samples are included in each class, whereas 150 samples are included in the test dataset for each class. A sample is composed of 1024-point vibration sequences without overlap.

In this study, 8 different types of real failure data, as well as health data, were selected for experiments. The details of used data and descriptions of faults are shown in Table 4 and Table 5, respectively.

Similarly, Figure 7 depicts the fault classification results of the KFDA model with different kernel widths. It can be found that KFDA can achieve the best performance while the kernel width is selected as 610. For RFDA, the optimal parameters s,m are selected as s=450,m=100 as displayed in Figure 8. The number of retained latent variables for all methods is set as 8.

Figure 9 is used to display the projections onto the first three FDA loading vectors by standard FDA, KFDA and RFDA. As plotted in Figure 9, it can be observed that the distances between the clusters of different classes are closer. Thus, the fault diagnosis performance will degrade. The reason is that the magnitudes of faults in the PU datasets are smaller. The vibration signals contain more noise. However, the clusters derived by RFDA and KFDA have larger discrepancy than FDA. Thus, higher accuracy rates can be offered. Nevertheless, KFDA and RFDA can provide better discriminant ability than FDA.

Table 6 offers the accuracy rates and computation times of FDA, KFDA, and RFDA using the testing dataset. From the data in Table 6, the accuracy rates of FDA and KFDA are below 90%. RFDA can provide a slightly higher accuracy rate (i.e., 90.05%) than KFDA. However, KFDA and RFDA can derive better fault diagnosis performance than FDA. For computation time, FDA requires the lowest computation burden compared to KFDA and RFDA. As demonstrated in the case study of PU, the computation time of RFDA is close to that of the FDA. The average computation time of RFDA is 1.43 s. Due to the need for calculating the kernel matrix, the average computation time of KFDA (i.e., 3.24 s) is longer than that of FDA and RFDA.

To investigate the performance of the proposed RFDA based fault diagnostic system further, the confusion matrices are plotted in Figure 10. From the data in Figure 10, values in the black squares on the diagonal indicate the classification accuracy, while values in the other gray-white squares indicate the error rate of the corresponding sample type. Most of the samples are correctly classified. However, it can be seen that the samples collected under Fault 6 are misclassified as Fault 5 for all methods. Among the comparable methods, fewer samples of Fault 6 are misclassified using RFDA. Overall, the RFDA-based method can offer satisfying results in the field of computation time and accuracy rate.

## 5. Conclusions

This paper proposed a randomized Fisher discriminant analysis (RFDA) algorithm to diagnose the bearing fault. First, 12 time-domain feature sets were constructed by extracting the original vibration signals. Then the RFDA model was established for fault diagnosis by introducing a random Fourier feature. To classify the class of vibration data, Bayesian inference was applied. Experiments on the PU and CWRU datasets were conducted to assess the performance of the RFDA-based fault diagnosis, compared with FDA and KFDA methods. Results verified that the RFDA-based method can provide better fault diagnosis performance using less computation time.

Despite this, there are still some studies to continue in future work. First, the performance can be improved by the reduction of similar cases (i.e., defects for the same part) for the PU dataset. The exploration of manifold structure in the bearing dataset would be another alternative way to improve the RFDA methodology. Additionally, the diagnostic performance is influenced by the magnitudes of different defects. The level of defect should be taken into consideration. The incipient fault detection and diagnosis is another hot topic in the bearing fault diagnosis community. There are two types of choices of random Fourier features [33]. It would be interesting to compare and study the performance of different forms of random Fourier features in the fault diagnosis. 

## Figures and Tables

**Figure 1 sensors-22-08093-f001:**
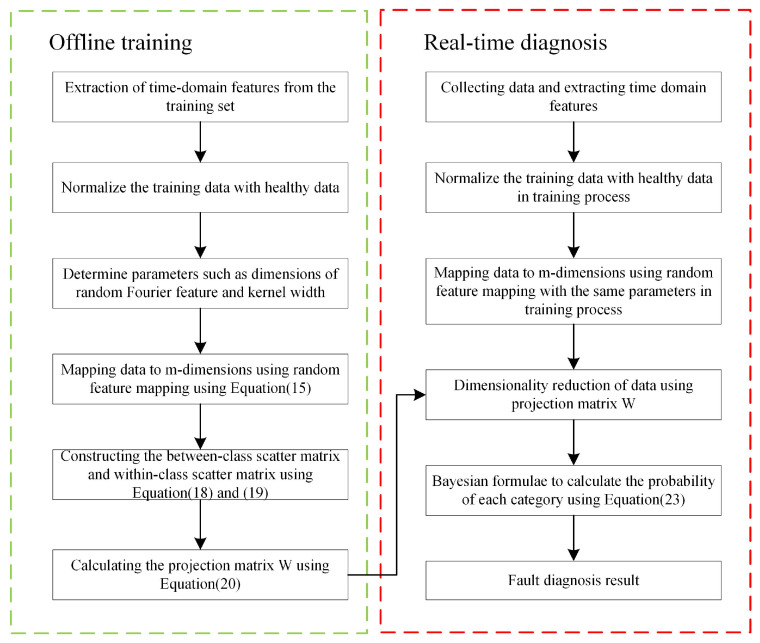
Flowchart of the proposed RFDA-based bearing diagnosis.

**Figure 2 sensors-22-08093-f002:**
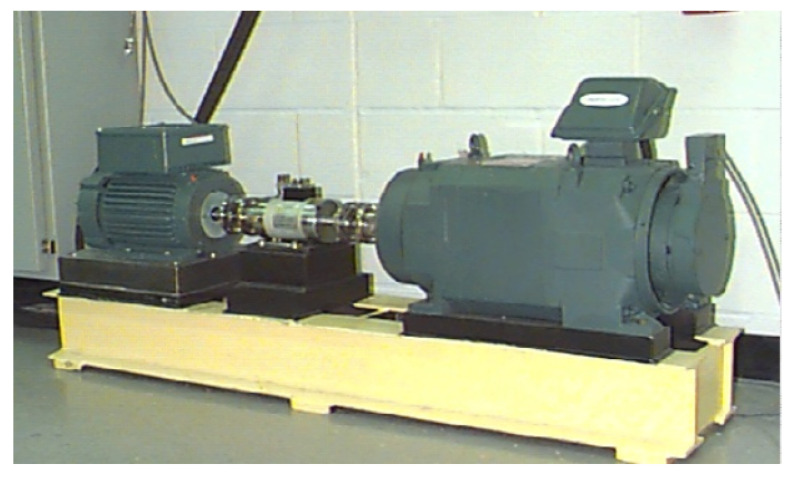
The experimental test rig of CWRU dataset.

**Figure 3 sensors-22-08093-f003:**
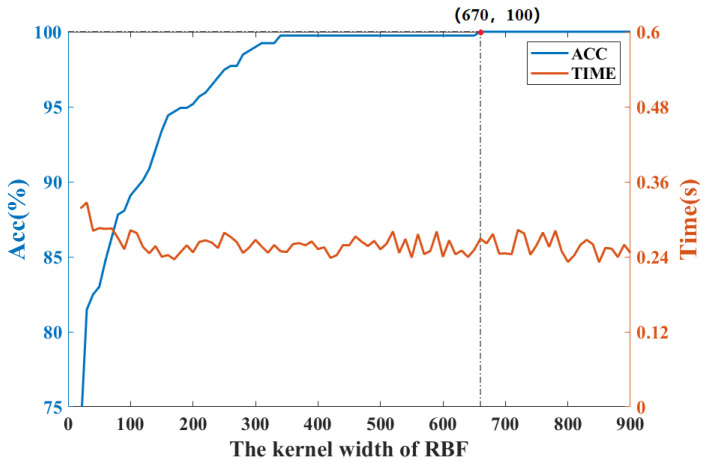
Accuracy rate and computation time vs. kernel width of RBF: CWRU dataset.

**Figure 4 sensors-22-08093-f004:**
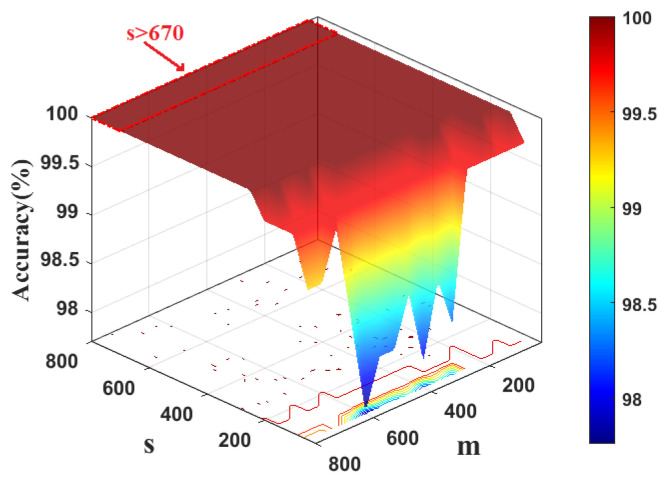
Computation time and accuracy rate vs. kernel width of RBF and dimensionality of random Fourier feature: CWRU dataset.

**Figure 5 sensors-22-08093-f005:**
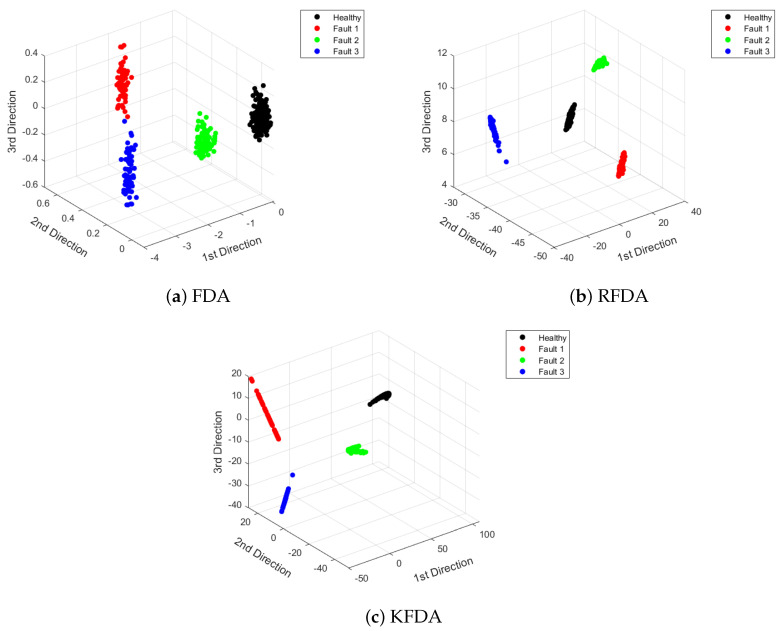
Visualization of dimensionality reduction: CWRU dataset.

**Figure 6 sensors-22-08093-f006:**
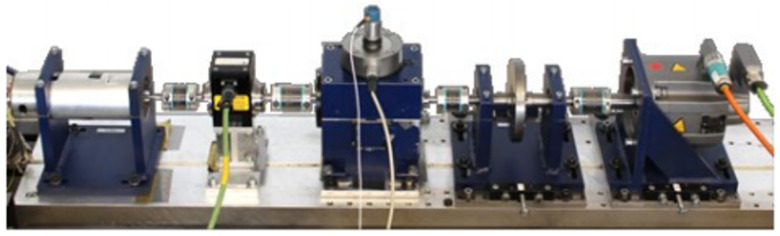
The experimental test rig of PU dataset test rig.

**Figure 7 sensors-22-08093-f007:**
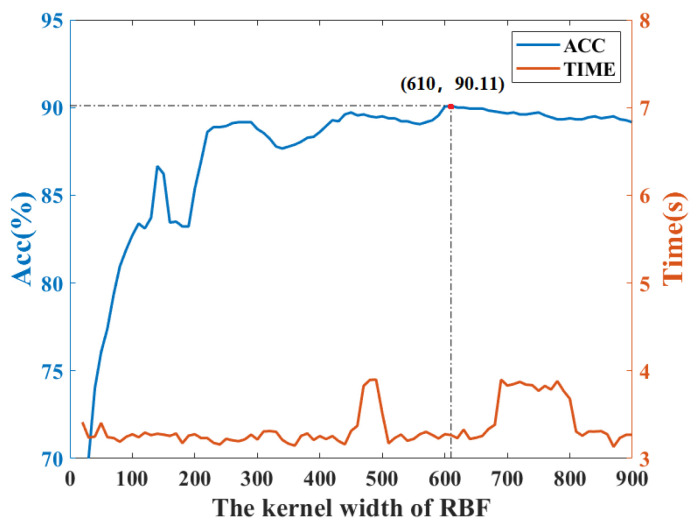
Computation time and accuracy rate vs. kernel width of RBF: PU dataset.

**Figure 8 sensors-22-08093-f008:**
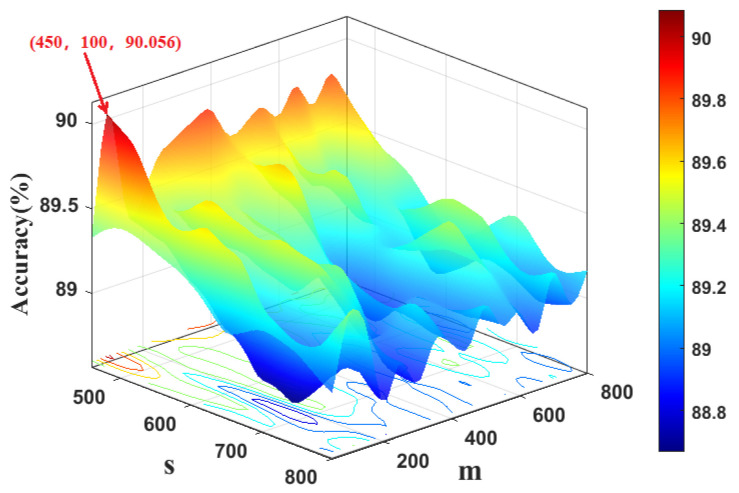
Computation time and accuracy rate vs. kernel width of RBF and dimensionality of random Fourier feature: PU dataset.

**Figure 9 sensors-22-08093-f009:**
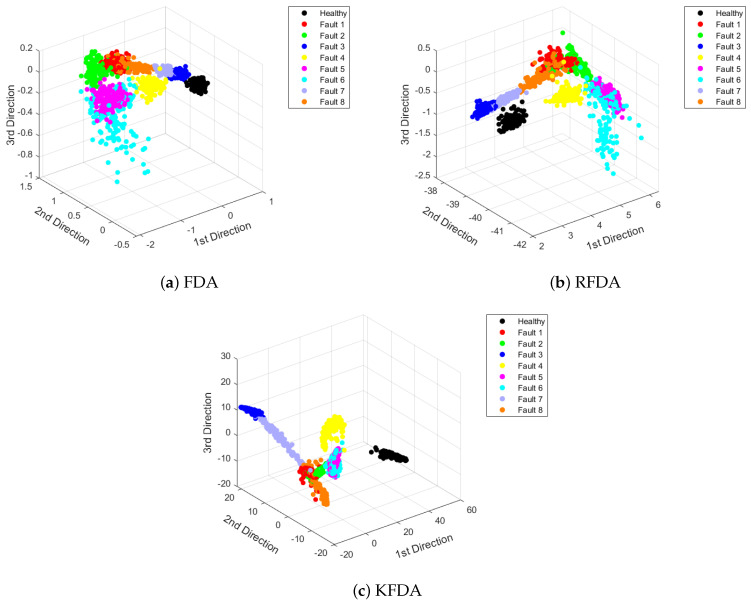
Visualization of dimensionality reduction: PU dataset.

**Figure 10 sensors-22-08093-f010:**
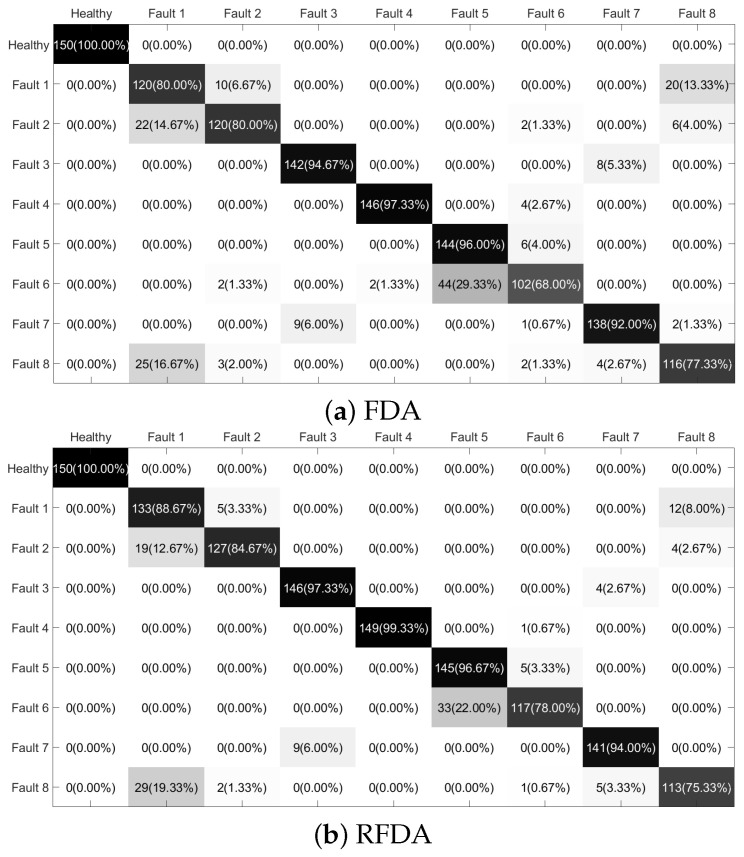
Confusion matrices: PU dataset.

**Table 1 sensors-22-08093-t001:** The 12 time-domain features.

Features	Equations
Peak	$x_{p}=\max\left(x\right)$
Peak-to-peak	$x_{pp}=\max\left(x\right)-\min\left(x\right)$
Mean	$\bar{x}=\frac{\sum_{i=1}^{n}x_{i}}{n}$
Absolute mean amplitude	$\bar{\left|x\right|}\frac{\sum_{i=1}^{n}\left|x_{i}\right|}{n}$
Square root amplitude	$x_{sra}={\left(\frac{\sqrt{\sum_{i=1}^{n}\left|x_{i}\right|}}{n}\right)}^{2}$
Variance	$x_{v}=\frac{\sum_{i=1}^{n}\left(x_{i}-\bar{x}\right)}{n}$
Standard deviation	$x_{std}=\sqrt{\frac{\sum_{i=1}^{n}{\left(x_{i}-\bar{x}\right)}^{2}}{n}}$
Root mean square	$x_{rms}=\sqrt{\frac{\sum_{i=1}^{n}x_{i}^{2}}{n}}$
Kurtosis	$x_{k}=\frac{\sum_{i=1}^{n}{\left(x_{i}-\bar{x}\right)}^{4}}{\left(n-1\right)x_{std}^{4}}$
Skewness	$x_{ske}=\frac{\sum_{i=1}^{n}{\left(x_{i}-\bar{x}\right)}^{3}}{\left(n-1\right)x_{std}^{3}}$
Peak factor	$x_{pf}=\frac{x_{p}}{x_{rms}}$
Impulsive factor	$x_{if}=\frac{x_{p}}{\bar{\left|x\right|}}$

where $x=\{x_{1},x_{2},\cdots ,x_{n}\}$ denotes the row signal series of each sample, *n* is the corresponding length of the
vibration signal. In this study, the sample of vibration signals contains 1024 data points.

**Table 2 sensors-22-08093-t002:** The number of training and testing datasets and the descriptions of fault scenarios: CWRU dataset.

Bearing State	Fault Location	Train Number	Test Number	Characteristic Frequency (Hz)
Health	/	50	188	29.95
Fault 1	inner ring	50	68	162.1852
Fault 2	rolling element	50	69	141.0907667
Fault 3	outer ring	50	69	107.305

All the vibration signals were collected under the same motor loads at 1797 rpm and 0 HP.

**Table 3 sensors-22-08093-t003:** Computation times and accuracy rates using FDA, KFDA and RFDA: CWRU.

Method	Mean Acc	Mean Time (s)
FDA	100%	0.1146
KFDA	100%	0.2647
RFDA	100%	0.1232

**Table 4 sensors-22-08093-t004:** The number of training and testing datasets and the descriptions of fault scenarios: PU dataset.

Bearing State	Bearing Code	Train Number	Test Number
Health	K004	100	150
Fault 1	KA04	100	150
Fault 2	KA16	100	150
Fault 3	KA22	100	150
Fault 4	KA30	100	150
Fault 5	KB23	100	150
Fault 6	KB24	100	150
Fault 7	KB27	100	150
Fault 8	KI16	100	150

**Table 5 sensors-22-08093-t005:** Detailed description of PU datasets.

Bearing State	Bearing Code	Fault Position	Description
Health	K004	Healthy	Run-in period 5 h
Fault 1	KA04	Outer ring (SP, S, Level 1)	Caused by fatigue and pitting
Fault 2	KA16	Outer ring (SP, R, Level 2)	Caused by fatigue and pitting
Fault 3	KA22	Outer ring (SP, S, Level 1)	Caused by fatigue and pitting
Fault 4	KA30	Outer ring (D, R, Level 1)	Caused by plastic deform and indentation
Fault 5	KB23	Outer ring and inner ring (SP, M, Level 2)	Caused by fatigue and pitting
Fault 6	KB24	Outer ring and inner ring (D, M, Level 3)	Caused by fatigue and pitting
Fault 7	KB27	Outer ring and inner ring (D, M, Level 1)	Caused by plastic deform and indentation
Fault 8	KI16	Inner ring (SP, S, Level 1)	Caused by fatigue and pitting

SP—single point fault; S—single damage; D—distributed fault; R—repetitive damage; M—multiple damage

**Table 6 sensors-22-08093-t006:** Computation times and accuracy rates using FDA, KFDA and RFDA: PU.

Method	Mean Acc	Mean Time (s)
FDA	86.72%	1.35
KFDA	89.72%	3.24
RFDA	90.05%	1.43

## Data Availability

Not applicable.

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
