# Peer review of "Bearing Fault Diagnosis Based on Randomized Fisher Discriminant Analysis"

_sensors, 2022, doi:10.3390/s22218093_

Round 1
Reviewer 1 Report
I found your article very interesting, but in my opinion below remarks would improve your manuscript under the scientific level.
Comments and Suggestions for Authors:
1. In the Abstract I miss the most important information about the novelty and outcomes of conducted research. I recommend to substitute the description of the experiment and data analysis with above mentioned issues.
2. By the dimensionality reduction techniques I miss the recurrence analysis, please follow those two recommended papers to be cited:
· Ambrożkiewicz et al. (2022), The influence of the radial internal clearance on the dynamic response of self-aligning ball bearing. Mechanical Systems and Systems Processing, 171, 108954.
· Syta et al. (2021), Detection of cylinder misfire in an aircraft engine using linear and non-linear signal analysis. Measurement, 174, 108982.
3. In the end of Introduction, please provide at least two references referring to RFDA, there are no given scientific fundamentals to it.
4. There is misspelling in the word “remainder”, it is not “remained”!
5. How the normalization process of time-series is conducted, and what is the reason for doing it, when you refer to detection analysis?
6. I’m familiar with the data from CWRU and Paderborn University, so please fulfil the information data on studied bearings in Section 4. Moreover, please refer it to the characteristic frequency for each defect.
7. In results, you have shown the advantage by RFDA, however, 90% referring to the diagnostics is not such good. How could you improve the methodology? Maybe the reduction of similar cases (i.e. defects for the same part) in the analysis would be helpful?
8. In conclusions, you have shown the direction of future studies, but I suggest to consider the level of defect in every case.
Reviewer 2 Report
The paper proposes the use of randomized Fisher Discriminant Analysis for fault diagnosis of bearing raw vibration signals for which, at the best of the authors' knowledge, there is little investigation. In this sense, the paper presents a novel approach for bearing fault detection, based on a kernelized randomized version of the FDA method, that, to the authors' knowledge, had not been tested for this purpose in the literature.
Introduction is reasonably well explained and the authors succeed in introducing the topic to the reader. However, in one of the key parts of the approach, which relies in dimensionality reduction principles. As the authors say, the KFDA method "seeks to map the original data onto a high-dimensional feature space in which linearly separable feature space is expected for further classification", and methods "investigate the inherent manifold structure embedded in data". Being the key idea of the paper heavily based on the dimensionality reduction principles, I miss some discussion on other non-kernel methods, such as tSNE, UMAP, and obviously the deep autoencoder family of algorithms, such as the deep autoencoder, variational autoencoder, etc. and their convolutional versions, since the methods should analyze temporal information (vibration timeseries). All this should be properly connected to the main topic of the paper to strengthen it.
In linenumber 100, the authors refer the FDA as "one of the most powerful dimensionality reduction methods". I suppose the claim is out of context (maybe the authors pretended to refer the sentence to linear methods), since there are many recent DR methods that outperform the FDA.
Anyway, in my opinion the most critical part of the paper is the explanation of the random feature map. The content between lines 134 and 150 (and very specially, equation (9) and lines 137-140) is poorly explained. The authors rely on literature material, that is supposed to be found in some of the references (such as Bochner [24]); but due to the high relevance to the paper idea (this is the key of the method), and also since the audience can be expected to be broad, a much more clear explanation should be added, otherwise the mean reader will very probably fail to understand the point of the paper. In other words, in this very relevant an important part, the paper should be more self-contained. A thorough revision should be done here, specially to improve readability.
In equation (11) some symbols inside the cos() functions do not appear.
In Equation (16), I can not see the connection or the role within the next equations (17)(18) ... there seems to lack something
Lines 217 and 246, what are are "s" and "m"? where are they defined? it seems from equation (16) that "s" is the kernel width, but in line 212 it is called "c"... notation should be clarified
In Figure 9 all x, y, z, labels are missing for subfigures (a), (b) and (c). Despite this information is somewhat given in the text "the first 3 features on the feature space of 249 FDA, KFDA, and RFDA are plotted in Figure 9." , it is not sufficient, what features are those three? does the reader have to check the table 1 ?... this information should appear explicitly as labels in the figures.
In line 273 "The feature set of 12 time-domain is constructed" ... this sentence is incomplete (i would expect "12 time-domain [something] is constructed")
Despite the authors say several times (including the abstract) that "a bayesian classifier is employed" I have not seen any explanation or details on it ! This is a major issue
Round 2
Reviewer 1 Report
All remarks have been fulfilled. I recommend the paper for its publishing in the present form.
Reviewer 2 Report
The authors have improved the text in some of the comments done in the last review. Also the figures like figure (9) have been improved following the suggestions.
However, I have found the section corresponding to equations (8), (9) and (10) with serious shortcomings, both in terms of legibility and understandability as well as having some errors. This part is, at the same time, abstract, difficult to many readers (me included) but also very important, since it underpins the main concept used in the paper the randomized approach within the kernel.
Equation (8), despite it can be deduced, beta is not explicitly defined. This contributes to difficult the reader following the development.
"βω(xi)∗ is the complex conjugate of the inverse Fourier transform of βω(xk)."... this has to be wrong, I find it contradictory, the only difference between both terms is the congugation (not a Fourier inverse operation)
Equation (9), the "beta" is missing twice. Only the subindices w are included in the argument of the expectaction operator E[]
In line 158, the authors propose replacing exp(-jw^T(xi-xk)) by cos(w^T(xi-xk))... I got stuck here until, again, I read [33] where an explanation in terms of the positivity of the kernel and p(w) suggests that the real part (the cosine) of the exponential would remain... omitting this explanation is too much matter left to the reader
In equation (10), the authors move from exp() functions to cos() functions that have bias. Just two lines before, they were talking about substituting exp() to cos(w(xi-xk)) without bias... now without explanation they add a bias term. Once more, in [33] I understand that the bias is another alternative that "also" gives unbiased estimator of the expectation in (8). But as I understand it, it is another choice, not an equivalence.
